# Comparison of Phenol Adsorption Property and Mechanism onto Different Moroccan Clays

**Younes Dehmani** [1,*], **Dison S. P. Franco** [2], **Jordana Georgin** [2], **Taibi Lamhasni** [3], **Younes Brahmi** [4], **Rachid Oukhrib** [5], **Belfaquir Mustapha** [6], **Hamou Moussout** [6], **Hassan Ouallal** [7] **and Abouarnadasse Sadik** [1]

[1] Laboratory of Chemistry and Biology Applied to the Environment, Faculty of Sciences of Meknes, Moulay Ismail University, Meknes 50050, Morocco; abouarnadasse@yahhoo.fr

[2] Department of Civil and Environmental, Universidad de la Costa, CUC, Calle 58 #55-66, Barranquilla 50366, Colombia; francodison@gmail.com (D.S.P.F.); jordanageorgin89@gmail.com (J.G.)

[3] Institut National des Sciences de l'Archéologie et du Patrimoine (INSAP), BP 6828, Madinat al Irfane, Avenue Allal El Fassi, Angle rues 5 et 7, Rabat-Instituts, Rabat 10000, Morocco; t.lamhasni@gmail.com

[4] HTMR-Lab, Mohammed VI Polytechnic University (UM6P), Benguerir 43150, Morocco; younes.brahmi@um6p.ma

[5] Team of Physical Chemistry and Environment, Faculty of Sciences, IBN ZOHR University, Agadir 80000, Morocco; rachid.oukhrib@edu.uiz.ac.ma

[6] Laboratory of Advanced Materials and Process Engineering, Faculty of Sciences, University Ibn Tofail, PB: 133, Kenitra 14000, Morocco; moussouthammou@gmail.com (B.M.); mbelfaquir@gmail.com (H.M.)

[7] Laboratory of Physic-Chemistry Materials, Faculty of Sciences and Techniques, Department of Chemistry Fundamental and Applied, Moulay Ismaïl University, PO Bosc 509, Errachidia 52000, Morocco; hassanouallalaghbalou@gmail.com

\* Correspondence: dehmaniy@gmail.com; Tel.: +212-678937801

**Abstract:** This study focuses on the removal of phenol from aqueous media using Agouraï clay (Fes-Meknes-Morocco region) and Geulmima clay (Draa Tafilalet region). The characterization of the clay by Fourier Transform Infrared (FTIR) Spectroscopy, X-ray diffraction (XRD), N2 adsorption (BET), Scanning Electron Microscopy (SEM), and Thermogravimetric and differential thermal analysis (DTA/GTA) indicates that it is mainly composed of quartz, kaolinite, and illite. The results showed that raw Clay Agourai (RCA) and raw Clay Geulmima (RCG) adsorbed phenol very quickly and reached equilibrium after 30 min. Thermodynamic parameters reveal the physical nature of the adsorption, the spontaneity, and the sequence of the process. However, the structure and structural characterization of the solid before and after phenol adsorption indicated that the mechanism of the reaction was electrostatic and that hydrogen bonding played an important role in RCG, while kinetic modeling showed the pseudo-second-order model dynamics. The physics-statistics modeling was employed for describing the isotherm adsorption for both systems. It was found that the monolayer model with two different energy sites best describes adsorption irrespective of the system. The model indicates that the receptor density of each clay direct influences the adsorption capacity, demonstrating that the composition of the clay is the main source of the difference. Thermodynamic simulations have shown that the adsorption of phenol is spontaneous and endothermic, irrespective of the system. In addition, thermodynamic simulations show that the RCG could be adsorbed even further since the equilibrium was not achieved for any thermodynamic variable. The strength of this study lies in the determination of the adsorption mechanism of phenol on clay materials and the optimum values of temperature and pH.

**Keywords:** phenol adsorption; different clays; adsorption mechanism; physics-statistics modeling

## 1. Introduction

Water is the essential element of life, and the quality and quantity of water on earth ask critical questions of all social actors [1]. Water pollution by various organic and inorganic pollutants is a global problem that requires a solution in the source of pollutants and

a solution before releasing them into aquatic ecosystems [2]. The pollutants that alter water quality are various and diverse and especially include phenol and phenolic compounds widely used in several industrial and agricultural fields [3]. The remarkably high toxicity of phenol has prompted the services in charge of environmental protection to normalize its concentration in water to reduce its impacts on humans and the environment [4]. Despite the non-high concentrations of these compounds in the aquatic environment, their toxic effects and their impact on the environment are remarkable, and several studies in the literature show the effects of these pollutants on health and the environment [5]. There are numerous techniques for cleaning up water that has been contaminated by organic substances such as degradation, biological treatment, membrane treatment, and catalytic oxidation. However, adsorption is the technique of choice for the removal of phenol [6]. The major advantage of this technique is the possibility of using several types of adsorbent and the simplicity of the process [6]. Clays have proven to be very effective in removing phenol from wastewater [6]. Clay minerals are the most important materials because of their known properties of adsorption and retention of pollutants [7], as well as their ease of modification and/or functionalization [7].

Despite the huge potential of clays, Morocco is still behind in the use of these processes to produce new high-value clay materials that could lead to innovative applications [8–10]. Some studies have focused on adsorption on clays. Agnieszka Gładysz-Płaska used red clay to adsorb uranium [11], and the adsorption of cationic and anionic dyes on clays has been the subject of work by Islem Chaari and collaborators [12]. Meichen Wang was interested in the adsorption of benzo[a]pyrene on modified clays [13]. Several works on the adsorption of dyes on clays and modified clays are mentioned in the work of Abida Kausar [14]. The work of Ali Q. Selim and co-workers [15] aimed at adsorbing phosphates on chemically modified carbonaceous clays. Using a natural clay (kaolinite), Nouria Nabbou tried to remove fluoride from groundwater [16].

There is a special property that characterizes porous solids, especially clay. This characteristic is the ability to adsorb heavy metals and organic matter contained in aqueous solutions, as well as a cationic exchange capacity [15]. This arises mainly due to their natural acidity and their high specific surface. Its performance is affected by several parameters: temperature, pH, and the properties of the adsorbed elements. The Moroccan kaolin clays of the regions of Guelmima (Darra Tafilalet region) and Agourai (Fes Meknes region) are part of this class. They are currently used in the ceramic industry, but they are also potential candidates for use in pollution control.

It is in this general context that this study has as its main objective the valorization and comparison of two Moroccan clays, one from the region of Daraâ Tafilalet and one from the region of Fes Meknes, as natural adsorbents for the retention of phenol from an aqueous solution. This work is divided into two parts. The first part is devoted to the characterization of the raw material. In the second part, we present the adsorption experiments, the methodology adopted and the results obtained, and their interpretations. The general conclusion summarizes the main results of this research work.

## 2. Materials and Methods

### 2.1. Sampling Area

The clays used in this work come from the town of Agouraï, Fes-Meknes region, and the town of Geulmima, Draa Tafilalet region (Morocco), as shown on the map in Figure 1. The sampled clays were sieved, washed, and dried in the oven for one night at a temperature of 130 °C.

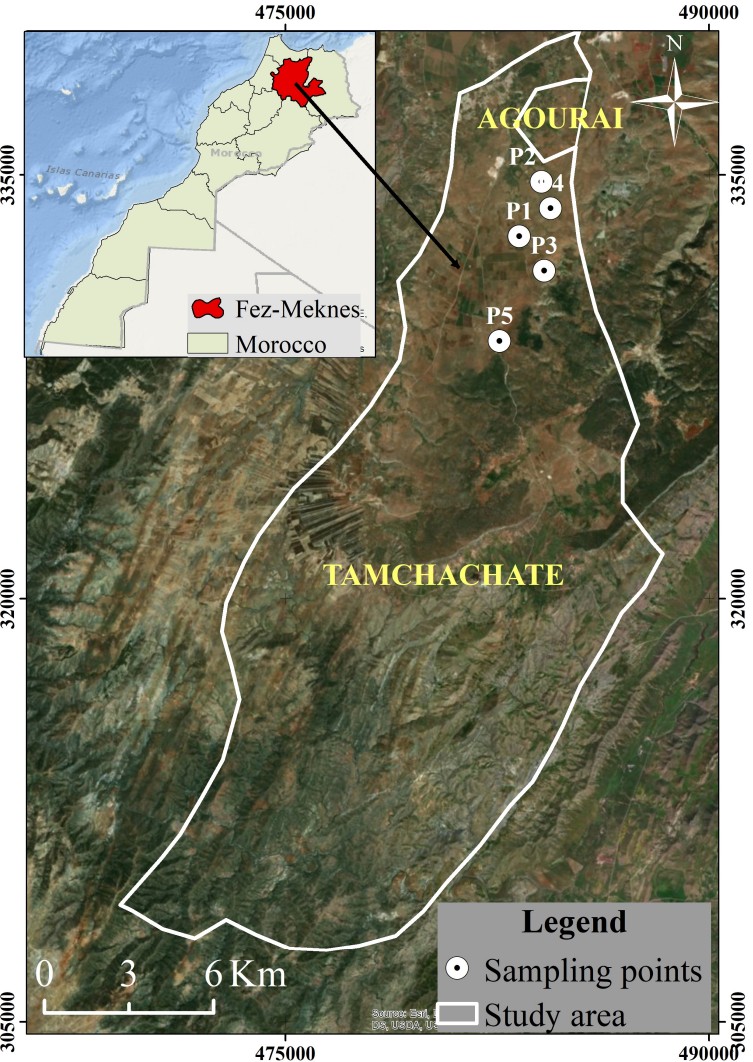

**Figure 1.** Geographical site of the clay used.

### 2.2. Adsorption Batch

The phenol adsorption experiments were conducted in accordance with the following protocol: a solution of phenol (20 mL) with a concentration that varied between 10 and 500 mg L$^{-1}$ was brought into contact with a mass of 0.1 g of solid at a constant temperature (T = 30, 40, and 50 °C) for pH = 4 and with a stirring rate of 600 rpm during the adsorption period. At the end of the assay, the mixtures were filtered and analyzed by UV/Vis spectroscopy. The residual concentration was determined based on a UV spectrometer calibration curve/Visible at λ = 270 nm (Shimadzu UV-1240). Equation (1) was used for the determination of the amount adsorbed:

$$Q_{ads} = \frac{(C_0 - C_e)}{m_{ads}} \times V_{sol} \qquad (1)$$

where $Q_{ads}$ is the adsorption capacity (mg g), $C_0$ is the initial concentration of the phenol (mg L), $C_e$ is the residual concentration of the phenol (mg L), $m_{ads}$ is the mass of adsorbent used (g), and $V_{sol}$ is the volume of the solution (L).

### 2.3. Kinetic Modelling

The experimental results were adjusted to nonlinear models in order to ascertain the adsorption mechanism of phenol on both solids (RCA and RCG). The modeling of

the adsorption kinetics was done using the pseudo-first-order and pseudo-second-order models [17]:

$$q_t = q_q(1 - e^{-tk_1})$$ (2)

$$q_t = q_e^2 k_2 \frac{t}{q_e k_2 t + 1}$$ (3)

where $q_t$ is the adsorption capacity according to time (mg g$^{-1}$), $q_e$ is the adsorption capacity at equilibrium (mg g$^{-1}$), $k_1$ is the pseudo-first-order kinetics constant (min$^{-1}$), $k_2$ is the pseudo-second-order kinetic constant (g mg$^{-1}$ min$^{-1}$), and t is the time (min).

*2.4. Isotherm Modeling through the Physic-Statistics Approach*

In this study, the adsorption of phenol onto several adsorbents was compared using a physical-statistical (Phys-Stat) modeling approach (RCA and RCG). The use of the grand canonical ensemble serves as the foundation for the Phys-stat models. which defines systems that, at a constant temperature, can exchange particles with their environment.

2.4.1. Monolayer Model with Single Energy Site (MLO)

The most commonly employed model is the monolayer model with one energy site (MLO). This model takes into consideration that a variable number of phenol molecules (n, dimensionless) is adsorbed in one type of receptor site with energy ($-\varepsilon_1$), reflecting directly in the quantity of adsorbed molecules, represented by the receptor density ($N_m$, mg g$^{-1}$), according to Equation (4) [18]:

$$q_e = \frac{nN_m}{1 + \left(\frac{C_{1/2}}{C_e}\right)^n} = \frac{q_m}{1 + \left(\frac{C_{1/2}}{C_e}\right)^n}$$ (4)

where $C_{1/2}$ is the concentration at half-saturation (mg L$^{-1}$) and $q_m$ is the adsorption capacity at saturation (mg g$^{-1}$).

2.4.2. Monolayer Model with Two Energy Sites (MLT)

When considering that a variable number of phenol molecules can be adsorbed onto two energetically different sites ($-\varepsilon_1$ and $-\varepsilon_2$) the monolayer model with two energy sites (MLT) is obtained. In this case, a different number of phenol molecules ($n_1$ and $n_2$, dimensionless) are adsorbed onto different receptors sites with different densities ($N_{m1}$ and $N_{m2}$, mg g$^{-1}$), according to Equation (5) [19]:

$$q_e = \frac{n_1 N_{m1}}{1 + \left(\frac{C_1}{C_e}\right)^{n_1}} + \frac{n_2 N_{m2}}{1 + \left(\frac{C_2}{C_e}\right)^{n_2}} = \frac{q_{m1}}{1 + \left(\frac{C_1}{C_e}\right)^{n_1}} + \frac{q_{m2}}{1 + \left(\frac{C_2}{C_e}\right)^{n_2}}$$ (5)

where $C_1$ and $C_2$ are the concentrations at half-saturation of the first and second receptor sites (mg L$^{-1}$), respectively, and $q_{m1}$ and $q_{m2}$ are the adsorption capacity at saturation of the first and second receptor sites (mg g$^{-1}$), respectively.

2.4.3. Double-Layer Model with One Energy Site (DLO)

Another possibility is the formation of dual layers. This hypothesis emerges from the consideration that the phenol molecules (n, dimensionless) can form a dual layer of molecules with a single energy of adsorption ($-\varepsilon_1$) with a single density of receptor site ($N_m$, mg g$^{-1}$) according to Equation (6) [20]:

$$q_e = nN_m \frac{\left(\frac{C_e}{C_{1/2}}\right)^n + 2\left(\frac{C_e}{C_{1/2}}\right)^{2n}}{1 + \left(\frac{C_e}{C_{1/2}}\right)^n + \left(\frac{C_e}{C_{1/2}}\right)^{2n}} = q_m \frac{\left(\frac{C_e}{C_{1/2}}\right)^n + 2\left(\frac{C_e}{C_{1/2}}\right)^{2n}}{1 + \left(\frac{C_e}{C_{1/2}}\right)^n + \left(\frac{C_e}{C_{1/2}}\right)^{2n}}$$ (6)

where $C_{1/2}$ (mg L$^{-1}$) is the concentration at half-saturation and $q_m$ is the adsorption capacity at saturation (mg g$^{-1}$).

### 2.4.4. Double-Layer Model with Two Energy Sites (DLT)

It is also possible that the phenol molecules (n, dimensionless) can form a double-layer, with each layer having different adsorption energy ($-\varepsilon_1$ and $-\varepsilon_2$), with the same receptor sites having the same density ($N_m$, mg g$^{-1}$), here presented by Equation (7) [21]:

$$q_e = nN_m \frac{\left(\frac{C_e}{C_1}\right)^n + 2\left(\frac{C_e}{C_2}\right)^{2n}}{1 + \left(\frac{C_e}{C_1}\right)^n + \left(\frac{C_e}{C_2}\right)^{2n}} = q_m \frac{\left(\frac{C_e}{C_1}\right)^n + 2\left(\frac{C_e}{C_2}\right)^{2n}}{1 + \left(\frac{C_e}{C_1}\right)^n + \left(\frac{C_e}{C_2}\right)^{2n}} \tag{7}$$

where $C_1$ and $C_2$ are the half-saturation concentration (mg L$^{-1}$), and $q_m$ is the adsorption capacity at saturation (mg g$^{-1}$).

### 2.4.5. Multilayer Model (MM)

Last, there is the possibility of forming a multilayer of adsorption. This model considers that the anchorage of the phenol molecules (n, dimensionless) depends on two energies, one corresponding to the first layer ($-\varepsilon_1$) and the other to all layers beyond the first ($-\varepsilon_2$), where the energy of adsorption of the first layer is always higher than the other layers. Similar to the other described model, the multilayer model considers only one density receptor site and the same number of molecules per layer [22]. The model is described according to the following Equations:

$$q_e = nN_m \left(\frac{F_1 + F_2 + F_3 + F_4}{G}\right) = q_m \left(\frac{F_1 + F_2 + F_3 + F_4}{G}\right) \tag{8}$$

$$F_1 = -\frac{2\left(\frac{C_e}{C_1}\right)^{2n}}{\left(1 - \left(\frac{C_e}{C_1}\right)^n\right)} + \frac{\left(\frac{C_e}{C_1}\right)^n \left(1 - \left(\frac{C_e}{C_1}\right)^n\right)}{\left(1 - \left(\frac{C_e}{C_1}\right)^n\right)^2} \tag{9}$$

$$F_2 = \frac{2\left(\frac{C_e}{C_1}\right)^n \left(\frac{C_e}{C_2}\right)^n \left(1 - \left(\frac{C_e}{C_2}\right)^{nN_2}\right)}{\left(1 - \left(\frac{C_e}{C_2}\right)^n\right)} \tag{10}$$

$$F_3 = \frac{\left(\frac{C_e}{C_1}\right)^n \left(\frac{C_e}{C_2}\right)^{2n} \left(\frac{C_e}{C_2}\right)^{nN_2} N_2}{\left(1 - \left(\frac{C_e}{C_2}\right)^n\right)} \tag{11}$$

$$F_4 = \frac{\left(\frac{C_e}{C_1}\right)^n \left(\frac{C_e}{C_2}\right)^{2n} \left(1 - \left(\frac{C_e}{C_2}\right)^{nN_2}\right)}{\left(1 - \left(\frac{C_e}{C_1}\right)^n\right)^2} \tag{12}$$

$$G = \frac{\left(1 - \left(\frac{C_e}{C_1}\right)^{2n}\right)}{\left(1 - \left(\frac{C_e}{C_1}\right)^n\right)} + \frac{\left(\frac{C_e}{C_1}\right)^n \left(\frac{C_e}{C_2}\right)^n \left(1 - \left(\frac{C_e}{C_2}\right)^{nN_2}\right)}{\left(1 - \left(\frac{C_e}{C_2}\right)^n\right)} \tag{13}$$

where $N_m$ is the density receptor of the receptor site (mg g$^{-1}$), $N_2$ is the number of formed layers, $C_1$ and $C_2$ are the half-saturation concentration (mg L$^{-1}$), and $q_m$ is the adsorption capacity at saturation (mg g$^{-1}$).

### 2.5. Parameter Estimation and Model Evaluation

The steric parameters were estimated using Matlab scripting programming. The built-in Matlab functions were utilized for this: particleswarm for the initial guess estimation, and lsqnonlin or nlinfit for incorrect particleswarm estimation. The difference between *lsqnonlin* and *nlinfit* is in the restrictions, where the first is used for a limited boundary and the second performs optimization for an undefined boundary. For the model evaluation, the coefficient of correlation ($R^2$), adjusted coefficient of determination ($R^2_{adj}$), minimum squared error (MSR, $(mg\ g^{-1})^2$), average relative error (ARE, %), and Bayesian Criterion Indicator (BIC) were employed [22]:

$$R^2 = 1 - \frac{\sum_{i=1}^{n}\left(q_{exp} - q_{pred}\right)^2}{\sum_{i=1}^{n}\left(q_{exp} - \bar{q}_{exp}\right)^2} \tag{14}$$

$$ARE = \frac{100\%}{n}\sum_{i=1}^{n}\left|\frac{q_{exp} - q_{pred}}{q_{exp}}\right| \tag{15}$$

$$MSR = \frac{1}{n-p}\sum_{i=1}^{n}\left(q_{exp} - q_{pred}\right)^2 \tag{16}$$

$$BIC = nLn\left(\frac{RSS}{n}\right) + pLn(n) \tag{17}$$

where $q_{exp}$ is the experimental adsorption capacity at the equilibrium ($mg\ g^{-1}$) and $q_{pred}$ is the predicted adsorption capacity at the equilibrium ($mg\ g^{-1}$), n is the number of experimental data points, and p is the number of parameters used in the model.

### 2.6. Structure Characterization

First, an "Axion" type X-ray fluorescence spectrometer with a dispersion of 1 kW wavelength was used to measure the X-ray fluorescence. The UATRS laboratory and CNRST in Rabat, Morocco, performed this chemical analysis. Then, using an X'PERT MPD-PRO wide-angle X-ray powder diffractometer equipped with a diffracted beam monochromator and Ni-filtered CuK radiation ($\lambda$ = 1.5406), X-ray diffraction (XRD) patterns were captured. With a counting period of 2.0 s and increments of 0.02°, the 2θ angle was scanned from 4° to 30°. Then, using a Fourier transform infrared spectrometer, the Fourier transforms infrared (FTIR) spectra of RCG and CCG were characterized (VERTEX70). The samples were made in a conventional manner on KBr discs from extremely well-dried mixes containing about 4% (*w/w*). FTIR spectra between 4000 and 400 cm$^{-1}$ were captured. Using LABSYS/Evo thermal, Thermogravimetric Analysis/Differential Thermal Analysis (TGA/DTA) studies were performed in an air environment. The samples were heated linearly (T = T0 + t) at a rate of 20 °C/min from ambient to 600 °C. In order to determine the textural characteristics, BET Nitrogen adsorption measurements were acquired using Micromeritics ASAP 2010. SEM images were acquired using a Quanta 200 EIF microscope equipped with standard secondary electron (SE) and backscattered electron (BSE) detectors. The electron beam is produced by a conventional tungsten electron source, which can resolve features as small as 3 nm under optimal operating conditions. The microscope is equipped with an Oxford Inca energy dispersive X-ray (EDX) system for elemental chemical analysis.

## 3. Results

### 3.1. X-ray Fluorescence

Our findings showed that, of the clays under study, silica, alumina, and calcium oxide made up around 66% of the composition of the RCG clays and 50 percent of the RCA clays, respectively. The concentrations of alkali and alkaline earth metals were lower in both clay solids (Table 1). This chemical analysis showed a 10.5% lower alumina content compared

to the alumina content of fire clay: 45% [15,23]. Moreover, the weight of calcite in the clay of the Fes Meknes region is higher than that of Geulmima. The presence of calcite reduces the shrinkage of the agglomerate.

**Table 1.** Chemical composition of the studied clays.

| Adsorbent | Oxide (Weight %) | | | | | | | | |
|---|---|---|---|---|---|---|---|---|---|
| | SiO$_2$ | Al$_2$O$_3$ | CaO | MgO | Fe$_2$O$_3$ | S | BaO | P$_2$O$_5$ | L.O.I |
| RCA | 38.74 | 10.60 | 16.80 | 0.92 | 6.00 | 0.19 | 0.02 | 0.10 | 16.61 |
| RCG | 48.39 | 16.16 | 2.17 | 1.75 | 15.44 | 3.55 | 0.26 | 1.39 | 10.09 |

### 3.2. X-ray Patterns of RCA and RCG

The diffractograms of RCG and RCA are shown in Figure 2. The diffractogram of RCG shows that Quartz (Q), Kaolinite (K), and Illite make up the majority of the clay (I). Conversely, the XRD diffractogram of RCA shows that the composition consists of kaolinite (Al$_2$Si$_2$O$_5$(OH)$_4$), illite [(K, H$_3$O) Al$_2$Si$_3$AlO$_{10}$(OH)$_2$], quartz (SiO$_2$), and calcite (CaCO$_3$). However, a comparison of the diffraction patterns shows the existence of two strong peaks, the first of which is connected to quartz and the other to calcite, indicating that this clay is heterogeneous [15,24,25].

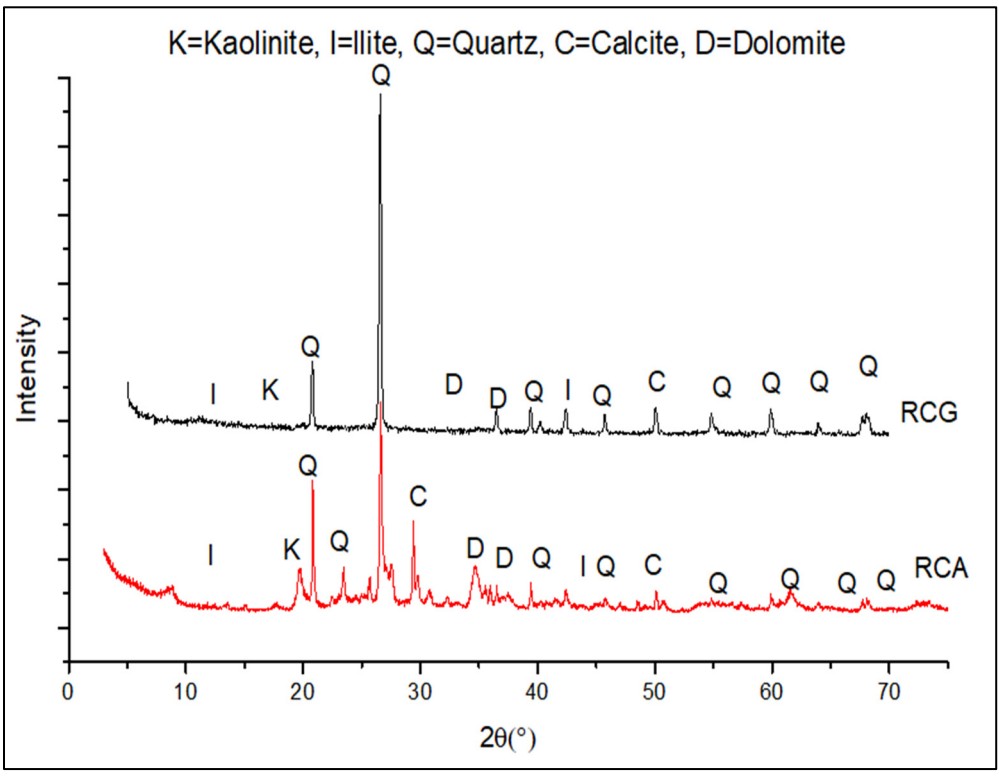

**Figure 2.** XRD patterns of RCA and RCG.

### 3.3. FTIR Spectra of RCA and RCG

Figure 3 shows the FTIR spectra of RCA and RCG. Both spectra show a broad absorption band around 3425 cm$^{-1}$ due to the stretching and bending vibrations, respectively, of adsorbed H$_2$O [26]. It is possible to assign the bands at 3649, 3690, and 3650 cm$^{-1}$ to the hydroxyl group's stretching vibration in various environments (Al, AlOH), (Al, MgOH), or (Al, FeOH) [27]. For the RCA clay, a band located at 1380 cm$^{-1}$ indicates the presence of calcite, while the bands detected at 2870, 1800, 1435, 870, and 715 cm$^{-1}$ are attributable to the deformation and elongation vibrations of calcite (CaCO$_3$) [28]. Vibrational distortion of the Si-O-Al bond is responsible for the band seen at 800 cm$^{-1}$. The bands at 470 and

520 cm$^{-1}$ are related to the deformation vibrations of the Si-O bond in quartz, while the band at 1030 cm$^{-1}$ is associated with the elongation vibration of the Si-O-Si bond in kaolinite or quartz [29]. The bands at 420 cm$^{-1}$, 470 cm$^{-1}$, and 525 cm$^{-1}$ correspond to the deformations of Si-O-Fe, Si-O-Mg, and Si-O-Al. Due to the Si-O stretching, a band seen at 1020 cm$^{-1}$ is appropriate [28]. The bands centered at 985, 836, 797, 674, and 508 cm$^{-1}$ can be attributed to the vibrational deformation of Al-OH-Al, Si-O-Al / Al-Mg-OH, cristobalite, Si-O-Mg, and Mg-OH bonds, respectively [30]. In addition to the Al-OH-Al strain vibration bonds, the band centered at 915 cm$^{-1}$ is also attributable to the presence of kaolinite. There are quartz-corresponding absorption bands at 797 and 779 cm$^{-1}$ [31].

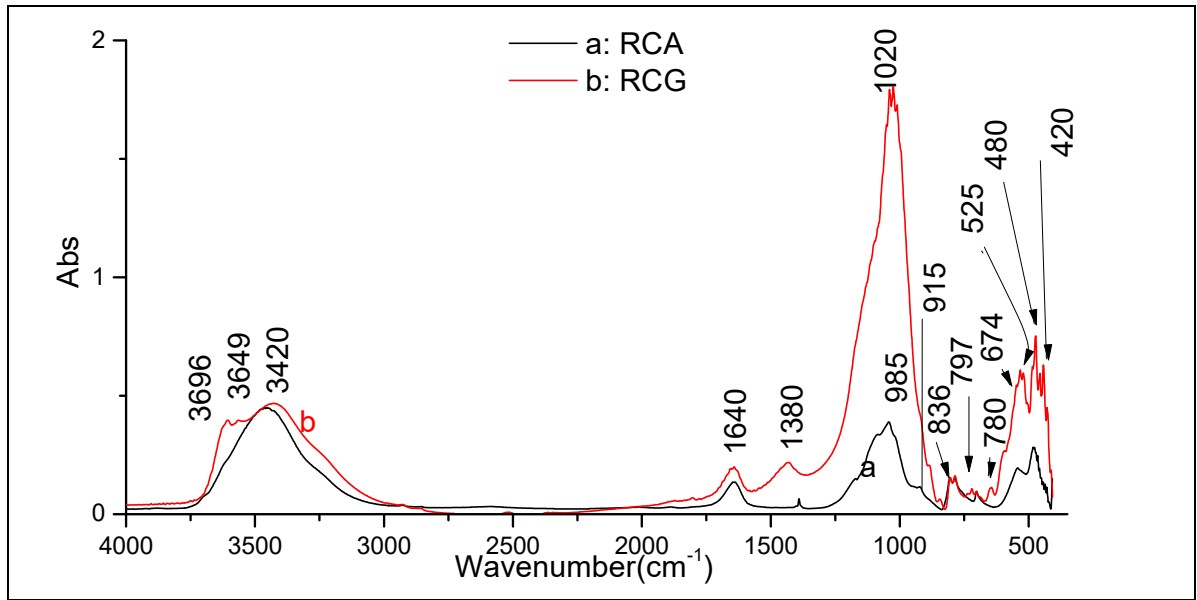

**Figure 3.** FTIR spectra of RCA and RCG.

### 3.4. TGA/TDA

Since both solids only experience a mass loss due to water adsorbed on their surfaces, the TGA curves (Figure 4) demonstrate the remarkable thermal stability of the two clays up to 600 °C. The DTG thermogram for the Geulmima clay (RCG) does, in fact, show the existence of an endothermic peak at 120 °C that corresponds to the dehydration and mass loss (1.29%) of the physisorbed water. The thermogram of Agourai clay (RCA) shows that the global mass measured is evaluated at 27% by an endothermic peak at 75 °C and 128 °C. This loss corresponds to the elimination of surface water. Thermal analysis reveals that the two clays under study vary in that there is an exothermic peak between 280 and 380 °C. Another endothermic peak between 450 and 550 °C, which is accompanied by a minor loss of mass, has been linked to the dehydroxylation of the raw clay for the RCA. This peak has been linked to the removal of organic materials [32,33].

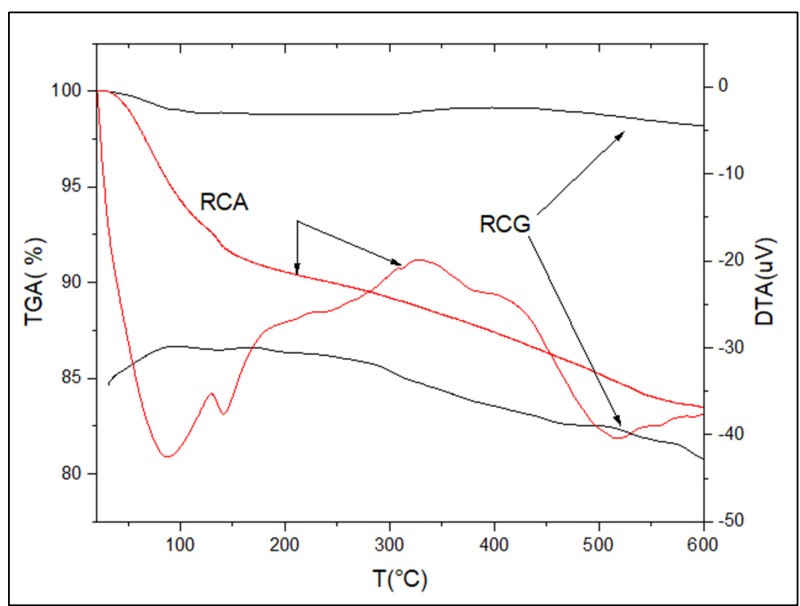

**Figure 4.** TGA/DTA Thermograms of RCA and RCG.

### 3.5. N₂ Adsorption/Desorption Isotherm of RCA and RCG

The nitrogen adsorption/desorption method (BET method by Breunuer, Emet, and Teller) is very important in the determination of the textural properties of solids. The textural parameters influence the catalytic and adsorptive capacity of the materials in the treatment of liquid pollutants. Based on the International Union of Pure and Applied Chemistry (IUPAC)'s recommended classification system for physical adsorption isotherms, the isotherms of the clays belong to the type IV isotherm (Figure 5) with a hysteresis loop of type $H_3$ [34]. There are platforms in the region where the P/P0 value is higher than the strongly increasing isotherms. These lines indicate that the samples contain macropores more than 50 nm in diameter [35]. Based on the data Table 2, the clay of the region of Fez Meknes (RCA) has a larger surface than the clay of Draa Tafilalet (RCG).even at the level of pore volume and pore diameter, the clay of Agourai is better than that of Geulmima

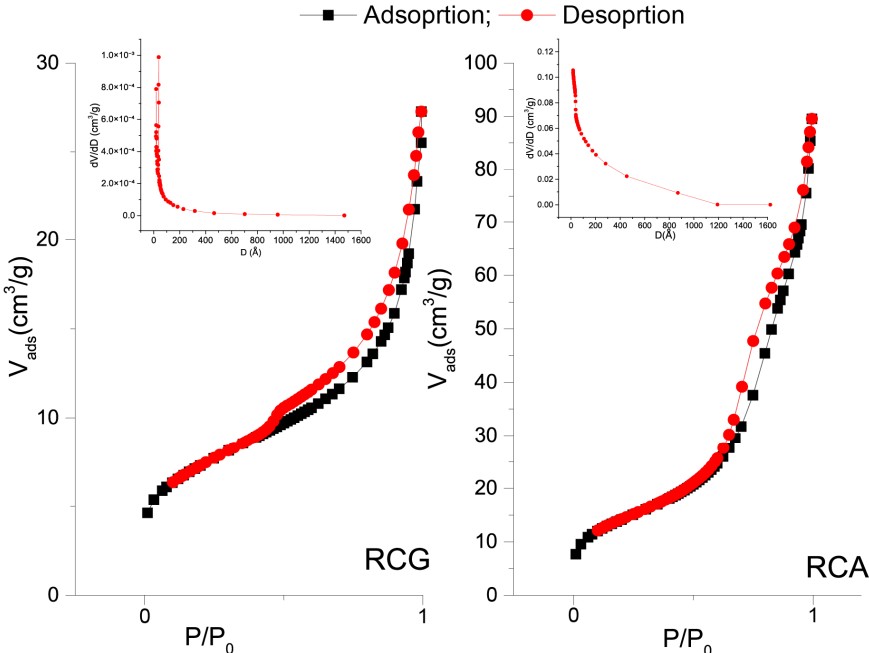

**Figure 5.** N₂ adsorption/desorption isotherms of RCA and RCG.

**Table 2.** Textural characteristics of RCA and ACA.

| Adsorbent | Proprieties | | |
| --- | --- | --- | --- |
| | Pore Volume (cm³ g⁻¹) | Specific Surface Area (m² g⁻¹) | Pore Diameter (Å) |
| RCA | 0.13 | 51.41 | 91.35 |
| RCG | 0.03 | 25.31 | 38.36 |

*3.6. SEM*

The SEM micrographic images (Figure 6) of both samples show the porous nature of our materials, which facilitates the penetration of the phenol molecules. Moreover, these images confirm that the structure of the clays is formed by a relatively homogeneous and compact clay matrix. The EDX spectra (Figure 7) show the presence of the main elements such as silica, aluminum, and oxygen, which confirms the validity of the X-ray fluorescence analysis.

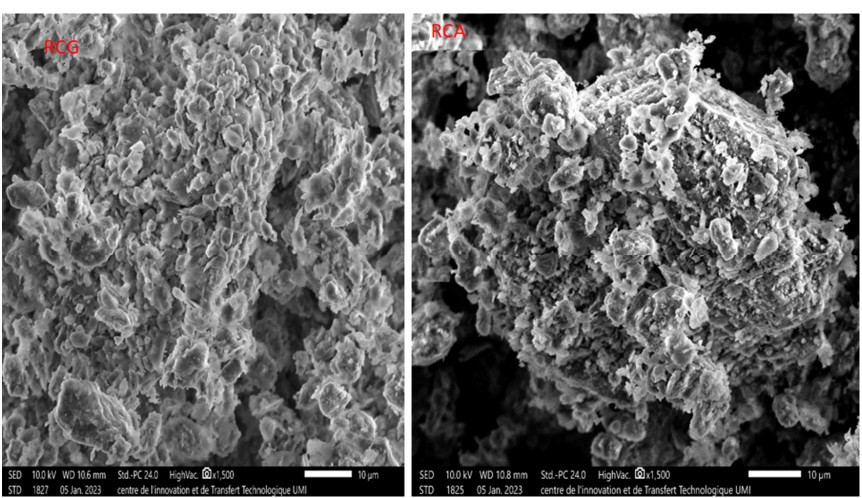

**Figure 6.** SEM images of RCA and RCG.

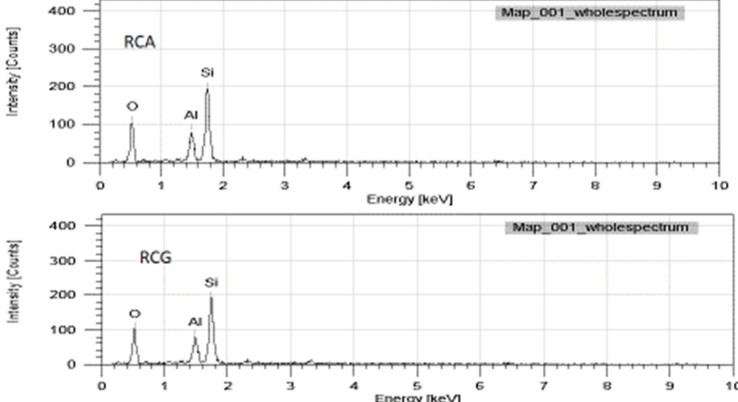

**Figure 7.** EDX data of RCA and RCG.

*3.7. Point of Zero Charge*

In the phenomenon of phenol adsorption, the determination of the point of zero charges is important because of the information on this point on the charge of the adsorbent surface and the intervals of change of this charge. The pH of zero charge point $pH_{pzc}$ of the RCG and RCA clays are 7.84 and 8.60, respectively (see Supplementary Materials Figures S1 and S2). Thus, for pH values higher than the pH of the points of null charge of the solids, the surface is negatively charged. At lower pH, the surface is positively charged. The adsorption of

phenol is more significant in the case of positively charged surfaces, so we work in the pH< pH$_{pzc}$ of the solids. A difference between the two solids at the pH$_{pzc}$ level can affect the adsorption capacity.

### 3.8. Determination of CEC

The cation exchange capacity was determined using the method of Delphine Aran [36]. Solids have a very high capacity compared to other types of clays [35]. The CEC of RCA is about 10.6 meq/100 g, while that of RCG is 16.64 meq/100 g. Compared with RCA, RCG has a higher switching capacity. This difference is evidenced by the percentages of elements that make up the two materials.

### 3.9. pH Effect

The pH of the solution is an important and specific factor in any liquid pollutant adsorption study as it affects the structure of the adsorbent and adsorbate as well as the adsorption mechanism. Therefore, it is logical to know the adsorption efficiency at different pH values to determine the optimal pH for adsorption. The literature gives two cases of the influence of pH on the adsorption of phenol [37]. We can see that in the acidic state, the positive charge on the surface of the adsorbent is dominant, so there is strong static electricity between the adsorbate and the adsorbent on the surface charge. In the ground state, the dramatic drop in adsorption capacity is evidenced by the nature of the predominantly positive charge on the solid surface [38].

For RCG, the adsorption capacity of phenol decreases with a slight increase in pH. The equilibrium adsorption capacity is 2.88 mg. g$^{-1}$ and 2.54 for pH = 4 and 11, respectively. The Agourai clay showed an important decrease in adsorption capacity from pH = 4 to pH = 11, from 1.2 to 0.8 mg g$^{-1}$, respectively (Figure 8). These results can be explained by the zero-charge pH parameter, which provides information about the dominant charge on the surface and can thus be demonstrated and interpreted in terms of the properties of the materials used. This can be explained by the fact that in the ground state (pH > pH$_{pcn}$), the predominant charge on the adsorbent surface is negative, which reduces the adsorption of the same charged phenoxide. In the acidic state, the positive charge on the surface of the adsorbent is dominant, so there is a relatively high electrostatic attraction between the positive charge on the surface of the adsorbent and the negative charge of the formed phenoxide, which promotes the adsorption. In short, heterogeneous composition leads to different interactions between adsorbate and adsorbent.

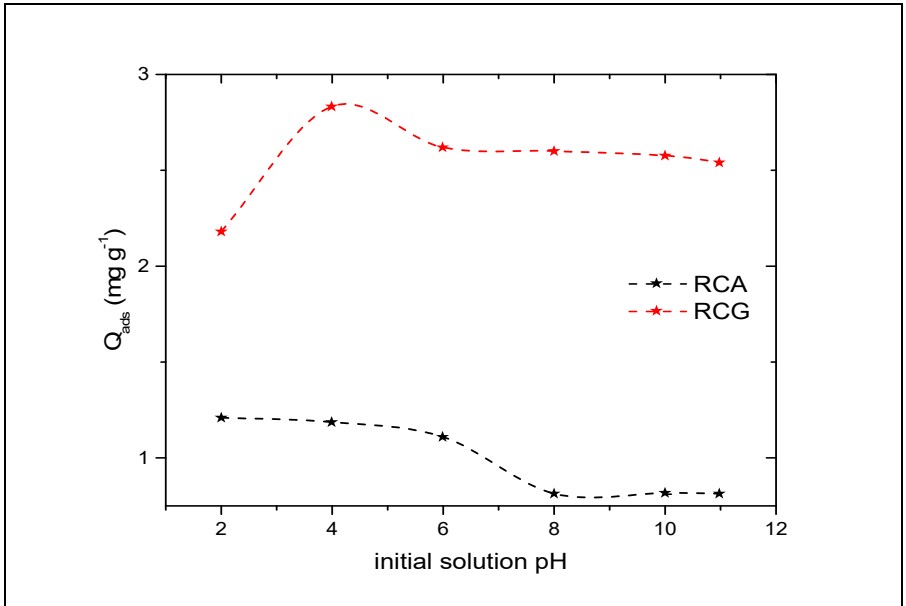

**Figure 8.** Effect of pH on the phenol adsorption onto RCA and RCG.

### 3.10. Adsorption of Phenol

### 3.10.1. Adsorption Kinetics

To evaluate the potential application of the samples in wastewater treatment, the variation in the adsorbed amounts of phenol on the two solids as a function of time and temperature is presented in Figure 9. The adsorbed amount in both solids increases rapidly up to 30 min, and then gradually increases up to 180 min. After this time, it remains almost unchanged at 360 min. The increase in the adsorbed quantity as a function of the temperature can be explained by the effect of the heat on the space between the particles of the solids, a dilatation of the latter imparts fast mobility to the phenol molecules, so an effect can be noticed on the molecules of the phenol by the increase in temperature, which accelerates the fixation on the surface of the clay materials. The amounts of phenol adsorbed by the RCG clay are 1.84, 2.43, and 3.52 mg g$^{-1}$ for temperatures 30 °C, 40 °C, and 50 °C, respectively. While the amount adsorbed by the RCA clay is 1.39, 2.10, and 2.71 mg g$^{-1}$, respectively, for the same temperatures. The adsorption capacity was better for RCG. This property is dependent on the structural and textural properties of this clay, particularly the zero charge point, which gives an important dominance of positive charges.

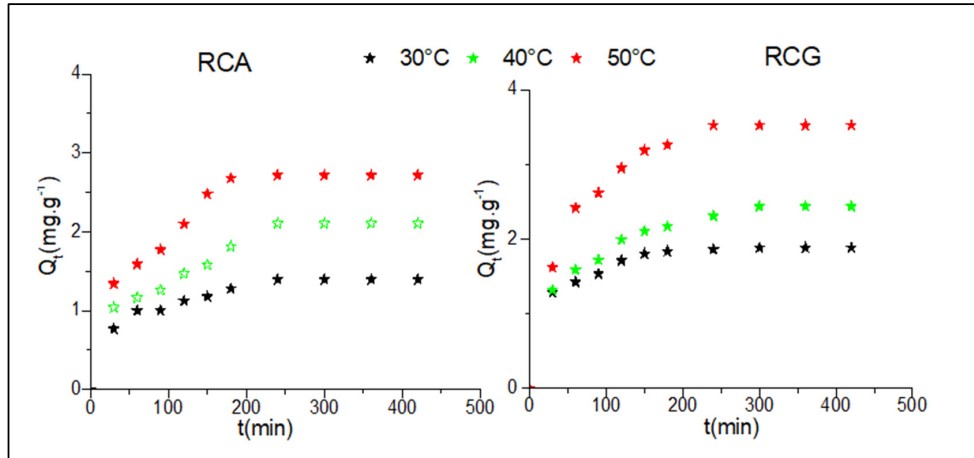

**Figure 9.** Adsorption kinetics of phenol onto RCA and RCG at different temperatures.

Understanding the adsorption kinetics is necessary to study adsorption because the process can predict the rate of adsorption and explain the mechanism of adsorption. Pseudo-first-order and pseudo-second-order kinetics were used to study the adsorption process of phenol on clay materials. Table 3 lists the parameters of the pseudo-first-order (Equation (2)) and pseudo-second-order (Equation (3)) model equations, and the model representations are shown in Figure 10 [39]. The results show that the adsorption kinetics are described by pseudo-second-order kinetics with a coefficient of determination close to 1 (Table 3).

**Table 3.** Kinetic parameters of linear and nonlinear modeling of phenol adsorption at different temperatures onto RCA and RCG.

| | | RCA | | | RCG | | |
|---|---|---|---|---|---|---|---|
| | **Parameters** | **Temperature (°C)** | | | **Temperature (°C)** | | |
| | | **30** | **40** | **50** | **30** | **40** | **50** |
| Models | $q_{texp}$ (mg g$^{-1}$) | 1.39 | 2.10 | 2.71 | 1.86 | 2.43 | 3.52 |
| Pseudo-first-order | $k_1$ (min$^{-1}$) | 0.02 | 0.01 | 0.14 | 0.03 | 0.01 | 0.01 |
| | $q_e$ (mg g$^{-1}$) | 1.34 | 2.10 | 3.73 | 1.83 | 2.34 | 3.48 |
| | $R^2$ | 0.95 | 0.93 | 0.96 | 0.96 | 0.96 | 0.99 |
| Pseudo-second-order | $k_2$ (g mg$^{-1}$.min$^{-1}$) | 0.101 | 0.2278 | 0.6124 | 0.4033 | 0.4918 | 1.521 |
| | $q_e$ (mg g$^{-1}$) | 1.515 | 2.481 | 3.17 | 1.992 | 2.663 | 3.99 |
| | $R^2$ | 0.9848 | 0.96018 | 0.9712 | 0.9911 | 0.9895 | 0.9955 |

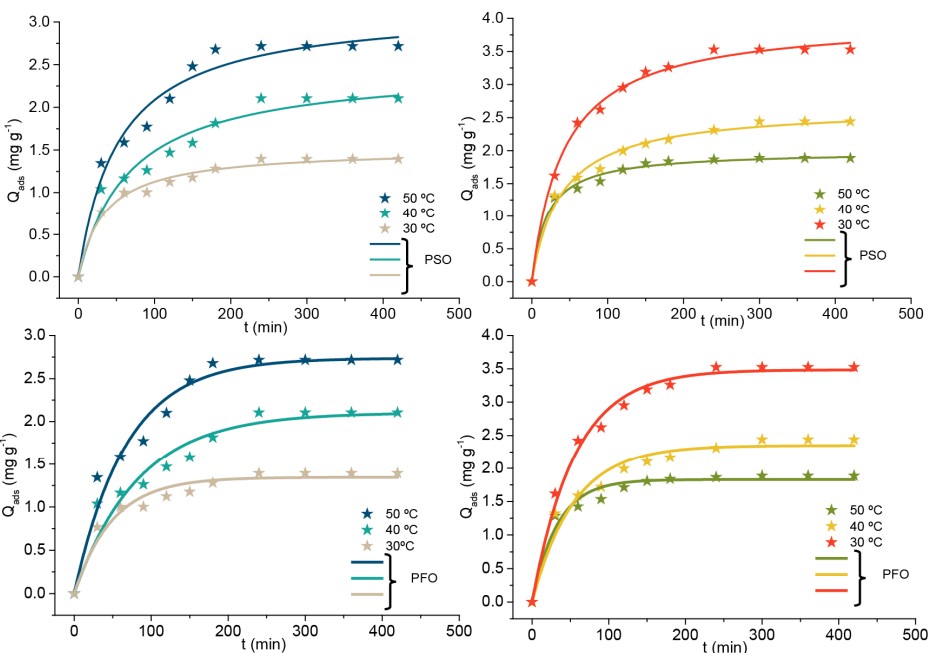

**Figure 10.** Adsorption kinetics with nonlinear models of pseudo-first-order and pseudo-second-order of phenol ($C0 = 5.10^{-4}$ M) onto RCA and RCG at pH = 4.

### 3.10.2. Adsorption Isotherms and Physical Statistical Interpretations

The experimental data and the physical statistical prediction are shown in Figure 11. Starting with the experimental data, it was found that both Moroccan clays (RCA and RCG) were able to adsorb the phenol, showing improvements according to the evolution of the system temperature. For the selection of the most suitable Phys-Stat model, the statistical indicators described in Section 2.6 and summarized in Table 4 were employed. Taking into consideration the results obtained from the statistical indicators, the first model to be eliminated is the MM, which corresponded to the Multilayer model. The failure to present a good correlation coefficient (R2), clearly indicates that the phenol molecules are not able to perform multilayer adsorption. The monolayer and dual-layer model present good statistical indicators. However, aiming to obtain the most adequate model, the BIC was used as an evaluation parameter. The BIC indicates that the monolayer model with two different energy sites (MLT) is the most adequate model for describing the adsorption of phenol onto the RCA and RCG clays. In addition to the statistical indicators, the steric parameters obtained for the other models, besides the MLT, do not present coherence. In other words, in cases where the concentration at half-saturation should be increasing, it diminishes or presents fluctuation. Overall the MLT indicates that the different number of phenol molecules ($n_1$ and $n_2$) are adsorbed onto two different energetic sites ($-\varepsilon_1$ and $-\varepsilon_2$), which leads to different receptor densities ($N_{m1}$ and $N_{m2}$). Thus, taking into consideration the MLT, the steric parameters are further discussed aiming to give insight into the phenol adsorption mechanism.

The number of adsorbed molecules per site is also known as the stereographic coefficient, which governs adsorption. The nature of the value also indicated how the phenol molecules are attached to the surface of the RCA and RCG. When the n values are below 1, the molecules are adsorbed in a parallel way. While, for values above 1, the molecules are attached on the surface in a perpendicular or non-parallel fashion [40]. In addition, the anchorage number ($n_a = 1/n$) represents the number of sites occupied by one molecule. The evolution of the number of adsorbed molecules per site according to the temperature and systems is presented in Figure 12. The first notable aspect is that the RCA and RCG clays have different numbers of molecules per site. For the RCA clay, the number of molecules per site is above 1, indicating that all molecules are adsorbed in a perpendicular way to the surface, irrespective of the receptor site. In addition, in all cases, the number of molecules

in site 1 is higher than in site 2 ($n_1 > n_2$), meaning that the phenol has a preference for the receptor site 1. In terms of anchor numbers, the estimated variation in phenol molecules is $0.02848 < n_{a1} < 0.03847$ for receptor site 1 and $0.1355 < n_{a2} < 0.6219$ for receptor site 2. This means that the phenol molecules occupy less than one full position in each case. This could be due to affinity or steric effects. In other words, the RCA clay has a deficit of receptor sites with a high affinity for the phenol molecules. The change in the number of molecules per site for the phenol/RCG is depicted in Figure 12B. In this case, the number of molecules per site is 10 times lower in comparison with the phenol/RCA. There are different possibilities to explain this difference between each material: (i) due to the density of receptor sites, (ii) due to textural proprieties, however, this could not be the reason due to the RCA having a higher specific surface area than the RCG, or (iii) the different compositions [41–43]. According to Table 1, the materials present differences in composition, with the RCG clay presenting higher quantities of silicon dioxide ($SiO_2$). Works in the literature reported that silicon dioxide directly influences the quantities of phenol adsorbed [6,44,45]. In a similar way, it is also possible to estimate the anchorage number for the phenol/RCG system; where $n_{a1}$ ranged from 0.8451 to 1.410 and 0.1824 to 0.8331 for $n_{a2}$, according to the temperature. This indicates that the phenol molecules tend to bind to more receptor sites of type one as the system temperature increases. As for receptor site type two, the phenol molecules tend to need less than one receptor site per molecule.

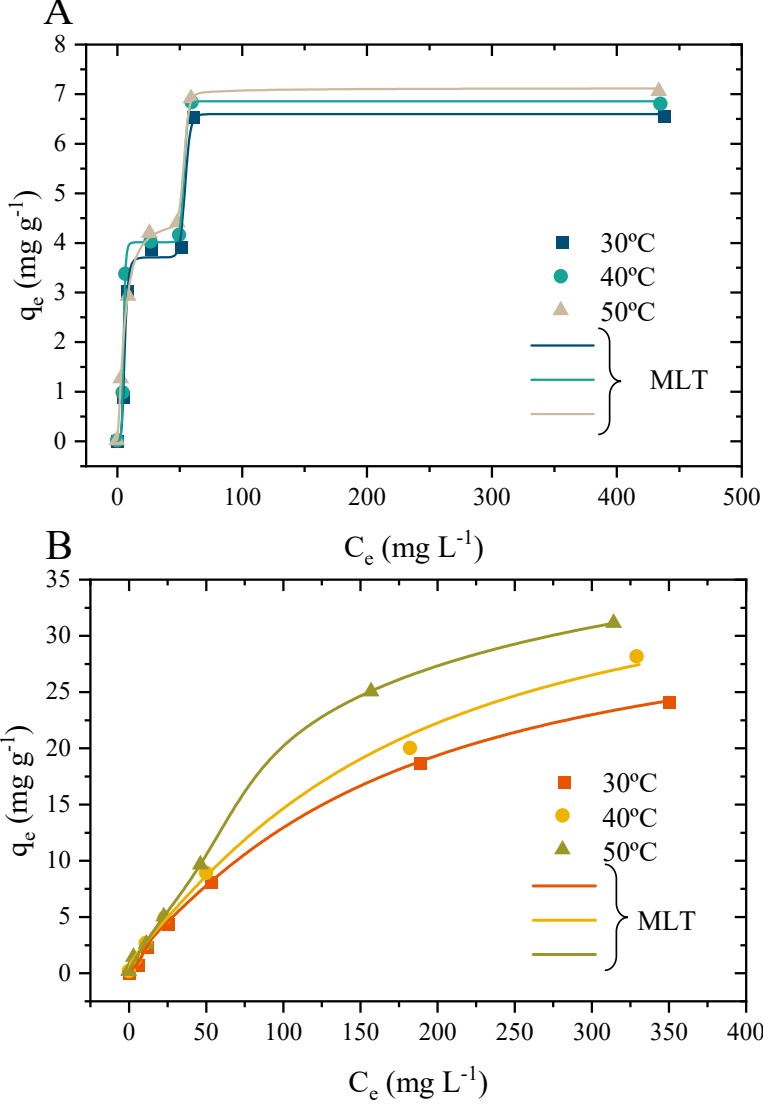

**Figure 11.** Phenol adsorption isotherms for the RCA (**A**) and RCG (**B**), lines are the model value predictions.

**Table 4.** Statistical indicators according to the model and adsorbent.

| Adsorbent | Model | T (°C) | Statistical Indicators | | | |
|---|---|---|---|---|---|---|
| | | | $R^2$ | ARE (%) | MSE $(mg\,g^{-1})^2$ | BIC |
| RCA | MLO | 20 | 0.8946 | 25.86 | 1.005 | 1.953 |
| | | 30 | 0.8766 | 29.38 | 1.2663 | 3.573 |
| | | 40 | 0.9371 | 10.50 | 0.6686 | −0.8976 |
| RCG | | 20 | 0.9942 | 11.74 | 0.7757 | 0.1421 |
| | | 30 | 0.9859 | 20.46 | 2.420 | 8.106 |
| | | 40 | 0.9967 | 22.55 | 0.7605 | 0.003962 |
| RCA | MLT | 20 | 0.9979 | 1.957 | 0.08037 | −19.59 |
| | | 30 | 0.9999 | 0.3973 | 0.00562 | −38.22 |
| | | 40 | 0.9995 | 1.613 | 0.02302 | −28.35 |
| RCG | | 20 | 0.9999 | 0.4556 | 0.008035 | −35.71 |
| | | 30 | 0.9999 | 3.943 | 0.09776 | −18.22 |
| | | 40 | 0.9998 | 5.033 | 0.1411 | −15.65 |
| RCA | DLO | 20 | 0.8939 | 26.19 | 1.011 | 2.000 |
| | | 30 | 0.8761 | 29.59 | 1.271 | 3.601 |
| | | 40 | 0.9369 | 10.90 | 0.6703 | −0.880 |
| RCG | | 20 | 0.9995 | 13.23 | 0.06972 | −16.72 |
| | | 30 | 0.9966 | 5.001 | 0.5792 | −1.902 |
| | | 40 | 0.9991 | 12.03 | 0.2102 | −8.998 |
| RCA | DLT | 20 | 0.8939 | 26.19 | 1.349 | 3.946 |
| | | 30 | 0.8761 | 29.59 | 1.695 | 5.547 |
| | | 40 | 0.9369 | 10.90 | 0.8937 | 1.066 |
| RCG | | 20 | 0.9995 | 13.23 | 0.09296 | −14.78 |
| | | 30 | 0.9966 | 5.001 | 0.7723 | 0.0442 |
| | | 40 | 0.9991 | 12.03 | 0.2802 | −7.052 |
| RCA | MM | 20 | 0.7985 | 28.92 | 3.840 | 10.38 |
| | | 30 | −0.2504 | 44.44 | 25.66 | 23.67 |
| | | 40 | −0.2296 | 28.12 | 26.14 | 23.80 |
| RCG | | 20 | −0.2536 | 83.57 | 334.9 | 41.66 |
| | | 30 | −0.3040 | 83.35 | 446.7 | 43.67 |
| | | 40 | −0.1736 | 83.41 | 541.9 | 45.03 |

The change in receptor density according to the evolution of the temperature and the system, phenol/RCA or phenol/RCG, is depicted in Figure 13. Starting with the RCA, it is possible that as the temperature of the system increases, the density of receptor two increases, while that of receptor one remains almost constant at around 0.5. This indicates that the affinity of receptor two is not dependent on the system temperature and that the increase in the adsorption capacity is due to receptor site two, the density of which increases. This effect can be related to two different causes: (i) a compensation regarding the number of molecules per site, in particular of receptor site one, where it achieved a higher value (above 10); (ii) the adsorbent material tends to change according to the temperature, facilitating the affinity between the phenol molecules and the RCA [46]. The density of the receptor sites for the phenol/RCG was found to observe more typical behavior, with the densities for both receptor sites tending to increase with the temperature. At all temperatures, the density of receptor site one ($N_{m1}$) is higher than that of receptor site two ($N_{m2}$), indicating that receptor one has a high affinity and thus, facilitates the adsorption of phenol. In addition, the minor decrease in the number of the adsorbed phenol molecules ($n_1$) direct reflects the behavior of the receptor density $N_{m1}$, where the value increases, indicating a higher number of molecules are attached to this site [47].

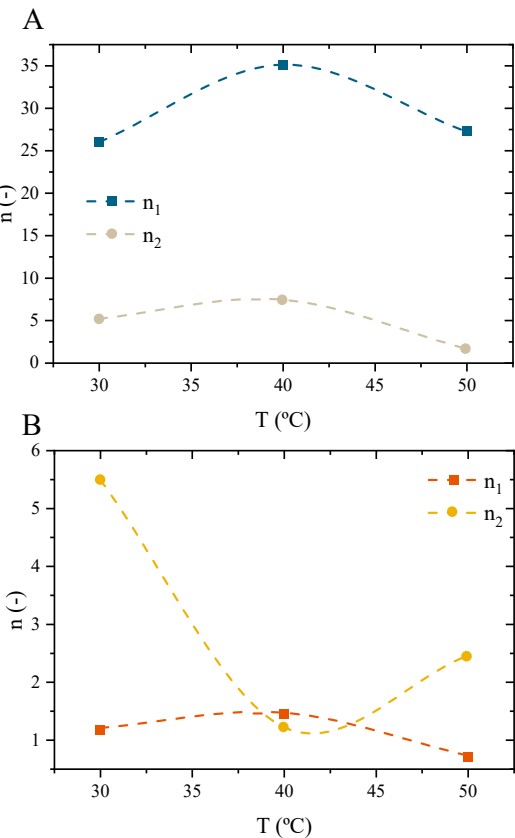

**Figure 12.** Evolution of the number of molecules per site parameters according to the temperature and system, RCA (**A**) and RCG (**B**).

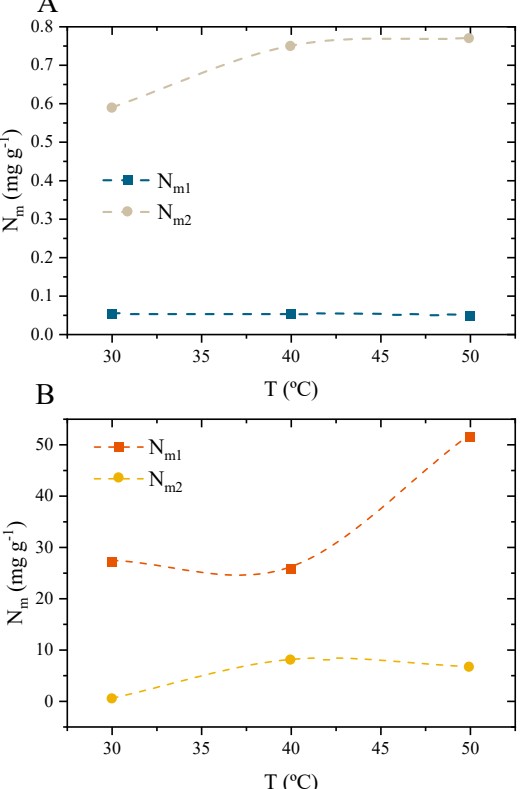

**Figure 13.** Receptor site density changes according to the temperature and system, RCA (**A**) and RCG (**B**).

From the half-concentration saturation parameters, it is possible to determine the adsorption energy according to the receptor site and system, as shown in Figure 14. The first aspect found is that the adsorption energy is positive for both systems, indicating endothermic adsorption based on physical interactions since the values were below 40 kJ mol$^{-1}$. For the phenol/RCA, it was found that receptor one has higher adsorption energy than receptor two, $\Delta E_1 > \Delta E_2$. In other words, receptor one has a higher affinity for the phenol. At first glance, this may seem to be in contradiction with the receptor densities, however, this may indicate that lower quantities of the receptor of type one may be present on the surface. For the phenol/RGC system, the energy of receptor one is also higher than receptor two. The minor difference is that, as the temperature evolves, a less them 1% decrease in the energy for the receptor site one occurs. The main explanation for this is the minor change that occurs in the number of molecules for receptor one [47]. In addition, these changes in the adsorption energy do not indicate a change in the nature of the adsorption, thus being endothermic for all the temperatures tackled in this study.

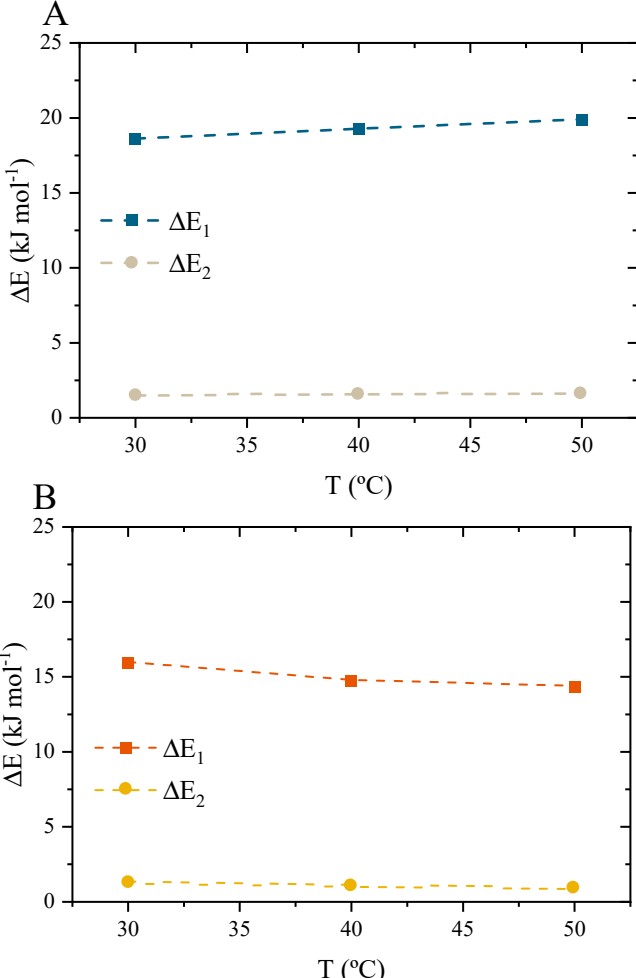

**Figure 14.** Estimated adsorption energy change according to the temperature and system, RCA (**A**) and RCG (**B**).

### 3.10.3. Application of the Stat-Phys Model with the Thermodynamic Potential Functions

Thermodynamic potential functions are quantities that are used to describe the state of a thermodynamic system and the energy exchanges that can occur within it. As it is possible to associate the grande canonical partition function to obtain the thermodynamic potential function of the configurational entropy ($S_A$, kJ mol$^{-1}$ K$^{-1}$), Gibbs free energy for

adsorption ($G_A$, kJ mol$^{-1}$), and internal energy of adsorption ($E_I$, kJ mol$^{-1}$. The definition of each thermodynamic variable is given by the following Equations [48]:

$$\frac{S_a}{k_B} = \ln\left(Z_{gc}\right) - \beta\frac{\partial}{\partial\beta}\ln\left(Z_{gc}\right) \tag{18}$$

$$G_a = \mu Q_0 \tag{19}$$

$$E_{int} = \frac{\mu}{\beta}\left(\frac{\partial}{\partial\mu}\ln\left(Z_{gc}\right)\right) - \frac{\partial}{\partial\beta}\ln\left(Z_{gc}\right) \tag{20}$$

where $Z_{gc}$ corresponds to the total grand canonical partition function, $\mu$ corresponds to the translational chemical potential of the phenol molecule (kJ mol$^{-1}$), $\beta$ corresponds to $1/k_B T$, $k_B$ corresponds to the Boltzmann constant, and T is the temperature of the system.

Taking into consideration that the MLT was the best Stat-Phys model to represent both systems, the thermodynamic parameters can be obtained according to Equations (21)–(23). The derivation of the Equations can be found elsewhere [47,49].

$$\frac{S_a}{k_B} = \begin{cases} N_{m1}\left[\ln\left(1+\left(\frac{C_e}{C_1}\right)^n\right) - \ln\left(1+\left(\frac{C_e}{C_1}\right)^n\right)\frac{\left(\frac{C_e}{C_1}\right)^n}{1+\left(\frac{C_e}{C_1}\right)^n}\right] \\ +N_{m2}\left[\ln\left(1+\left(\frac{C_e}{C_2}\right)^n\right) - \ln\left(1+\left(\frac{C_e}{C_2}\right)^n\right)\frac{\left(\frac{C_e}{C_2}\right)^n}{1+\left(\frac{C_e}{C_2}\right)^n}\right] \end{cases} \tag{21}$$

$$G_a\beta = \ln\left(\frac{C_e}{\left(\frac{2\pi m}{h^2\beta}\right)^{3/2}}\right)\left(\frac{Q_{m1}}{1+\left(\frac{C_1}{C_e}\right)^n} + \frac{Q_{m2}}{1+\left(\frac{C_2}{C_e}\right)^n}\right) \tag{22}$$

$$E_{int} = \begin{cases} N_{m1}\left[\mu\frac{\left(\frac{C_e}{C_1}\right)^n}{1+\left(\frac{C_e}{C_1}\right)^n}\frac{1}{\beta}\ln\left(1+\left(\frac{C_e}{C_1}\right)^n\right)\frac{\left(\frac{C_e}{C_1}\right)^n}{1+\left(\frac{C_e}{C_1}\right)^n}\right] \\ +N_{m2}\left[\mu\frac{\left(\frac{C_e}{C_2}\right)^n}{1+\left(\frac{C_e}{C_2}\right)^n} - \frac{1}{\beta}\ln\left(1+\left(\frac{C_e}{C_2}\right)^n\right)\frac{\left(\frac{C_e}{C_2}\right)^n}{1+\left(\frac{C_e}{C_2}\right)^n}\right] \end{cases} \tag{23}$$

### 3.10.4. Interpretations of the Potential Function

The simulations of the change in the configuration entropy according to the phenol equilibrium concentration and the systems are depicted in Figure 15. The simulations take into consideration the maximum concentration equilibrium obtained from the isotherms, meaning around 500 mg L$^{-1}$, for a fixed value. The first aspect to be observed is that the configuration entropy for the phenol/RCA presents two entropic peaks. The first is at around 5 to 7 mg L$^{-1}$, which corresponds to the concentration at half-saturation for receptor site two. After that, a second minor entropic peak occurs, corresponding to the receptor site one. After that, the entropic change diminishes until achieving equilibrium. The second aspect is regarding the phenol/RCG system, where, at first glance, the simulation does not show the entropic equilibrium and shows one single entropic peak for all temperatures. Regarding the entropic equilibrium, further simulation indicates that it is only reached after 1500 mg L$^{-1}$. This indicates that the adsorption could occur beyond the concentration investigated in this work. Regarding the single entropic peak, the main explanation is that receptor sites one and two present close half-saturation concentrations, meaning that the adsorption occurs at the same equilibrium concentration. Last, regarding the temperature effect, the results are in agreement with the endothermic nature, in which the temperature increases both the adsorption capacity and the entropic behavior.

The simulation of the Gibbs free energy according to the equilibrium concentration and temperature system is shown in Figure 16. The first observation is that the Gibbs free

energy is negative for all phenol equilibrium concentrations, temperatures, and systems, indicating that the adsorption of phenol is spontaneous at any given point. Regarding the different profiles of each simulation, the explanation goes hand in hand with the entropic behavior. In the first case, phenol/RCA displays a ladder-type profile. This profile is related to the different receptor sites' half-saturation concentrations, where site one is only adsorbed with high phenol concentrations. As for the phenol/RCG, the same effect of not presenting an equilibrium at this concentration indicates that the RCG can further adsorb phenol. The lack of a ladder-type profile also corroborates the entropic change, which presents a single peak, meaning that the adsorption of phenol is spontaneous and occurs in the same way for the two receptor sites.

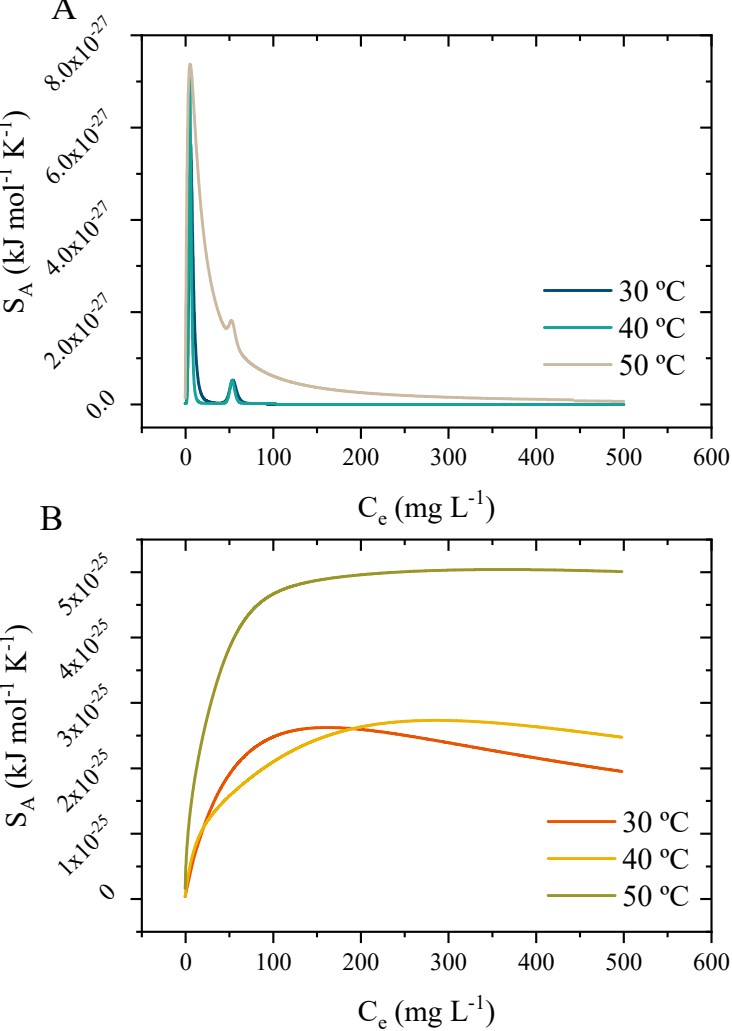

**Figure 15.** Entropic change according to the phenol equilibrium, temperature, and adsorption system, RCA (**A**) and RCG (**B**).

The simulations for the internal energy according to the phenol equilibrium concentration and system temperature are shown in Figure 17. As expected, the internal energy presents similar behavior to the simulations obtained for the configurational entropy and Gibbs free energy. In other words, the presence of ladders in the phenol/RCA system, due to the energic differences between the receptor sites and the lack of equilibrium for the phenol/RCG system, indicates that more phenol molecules could be adsorbed. Overall, the internal energy tends to increase with temperature, corroborating the results obtained for the experimental isotherms.

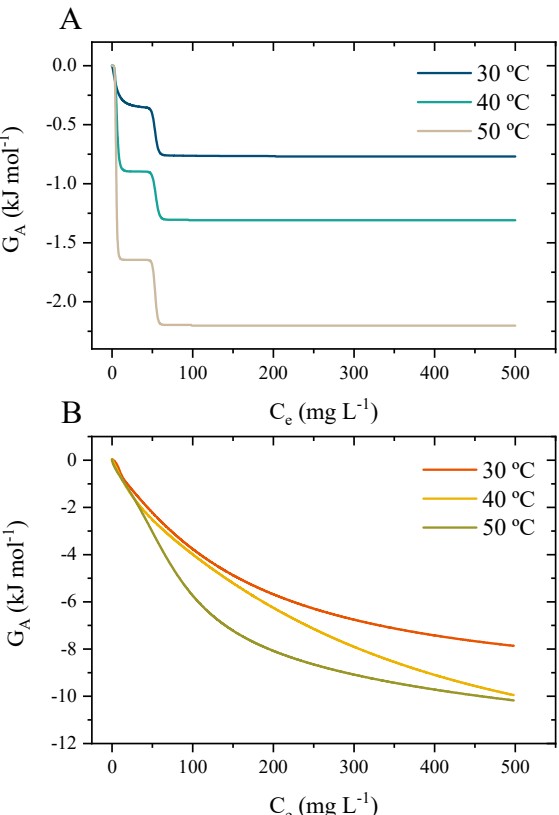

**Figure 16.** Gibbs free energy changes according to the phenol concentration, temperature, and system, RCA (**A**) and RCG (**B**).

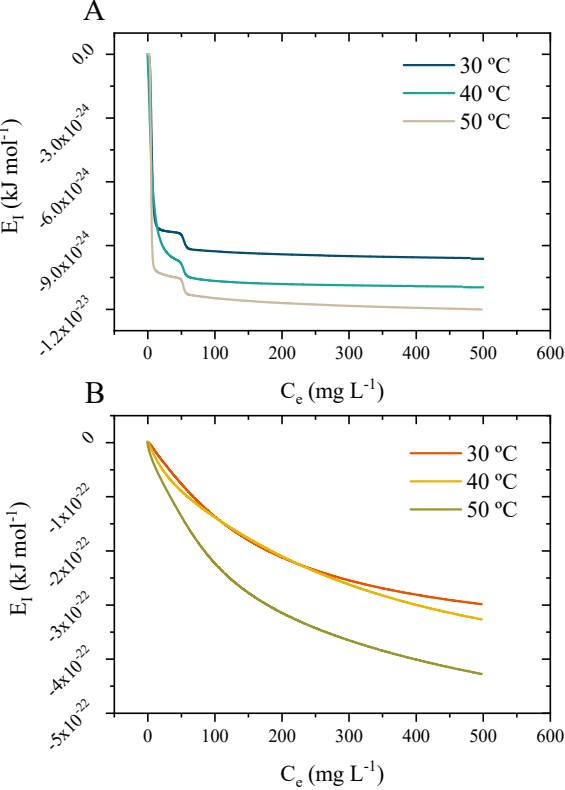

**Figure 17.** Internal energy changes according to the equilibrium concentration, temperature, and system, RCA (**A**) and RCG (**B**).

3.10.5. Mechanism of Phenol Adsorption

The adsorption process of phenol on clay usually relies on several physicochemical forces occurring at the solid-liquid interface, such as the interaction between phenol molecules and clay functional groups

➢ Van der Waals force: Dipole-dipole attraction between atoms or molecules through low-level electrical interference. This attractive force is very important for the adsorption of organic substances such as phenol. We are well aware of the inhomogeneity of clay solids.

➢ Coulomb force: The electrostatic force developed between a charged surface and an opposite charge. Surface charges can result from isostructural substitution or protonation or deprotonation of surface functional groups. Surface charge is determined by changes in the pH of the medium, and the pH at the point of zero charges can help us identify and assign the dominant charge on a clay surface.

➢ The hydrogen bonds, or the role of $H_2O$ in the adsorption, have intermolecular interactions that occur between hydrogen atoms and electronegative atoms (O, F, S, Cl).

In this work, we try to deepen the study of the mechanism of the adsorption of phenol on clay. For that, structural and textural characterization of the samples was used to determine the nature of the interactions between the adsorbate and the adsorbent either by FTIR or XRD (see the supplement of this work).

To better understand the adsorption mechanism of clay materials, the infrared spectra of phenol and clay were examined. Figures S3 and S5 (Supplementary Materials) show the FTIR spectra of RCG and RCA both before and after phenol adsorption. The internal OH units of the kaolinite structure and the water of hydration of sodium cations are responsible for the characteristic RCG and RCA bands at 3625 and 3420 $cm^{-1}$, respectively. [50]. With increasing phenol content, the intensity of the RCA and RCG bands (3400–3650 $cm^{-1}$) increases. These findings imply that phenol penetrates the kaolinite interlayer and forms a hydrogen connection with the water molecules in the cationic hydration spheres. In all samples, the water OH groups' bands at 1638 and 3467 $cm^{-1}$ appear first. Some bands developed in the spectra of the sample after phenol adsorption as opposed to the new peaks not seen in the Agourai clay samples before and after adsorption. While the band at 1584 $cm^{-1}$ is caused by C-O vibrations, the band at 1479 $cm^{-1}$ is caused by the stretching of the aromatic C=C bond. The CH band for Ph($10^{-3}$ M)/RCG is situated at 1450 $cm^{-1}$ in Figure 9. This alteration in the plane of the phenol CH group is brought on by bending vibrations. Additionally, vibrations below 1030 $cm^{-1}$ that correspond to the Si-O deformation and Si-O-Si stretching modes show band separation, indicating that they are involved in phenolic interactions [51]. Figures S2 and S6 (Supplementary Materials) displays the tilemaps for the RCG and CCG. The DRX analysis suggests that the interlobar space of the solid has been stripped as a result of the insertion of phenol into this space with a large dip angle, as seen by the removal of the quartz line at 20° in the spectrum in the RCG spectrum following phenol adsorption [45]. The distance from the quartz plane may only be extended if this adsorption mechanism is permitted; the average diameter of the phenol molecule is 5, depending on how it enters the interlayer gap. [15,52,53]. In addition, the maximum intensity in the RCG spectrum was reduced after phenol adsorption. In the clay samples collected in the Meknes region, the only thing missing was the lines in the raw clay attributed to calcite. In both diffractograms, there is no movement of lines or the appearance of new lines. Only one effect can be observed, and that is the reduction in peak intensity. After phenol adsorption, neither solid's spectrum displayed any new peaks, supporting the earlier finding of phenol's physical adsorption. According to the findings of XRD, FTIR, and SEM, surface adsorption rather than intercalation were the primary interactions between phenols and clay particles, which happened at the outer surface by electrostatic attraction.

## 4. Conclusions

The results of this laboratory-scale study demonstrate the utility of using RCA and RCG in the field of remediation of water bodies contaminated with organic pollutants. The adsorption kinetics of phenol on different adsorbents allowed the selection of Geulmima clay as the optimal adsorbent for phenol in an aqueous solution. The parameters under study, the pH value, and the temperature of the medium influence this kinetics. However, the structure and structural characterization of the solid before and after phenol adsorption indicated that the mechanism of the reaction was electrostatic and that hydrogen bonding played an important role in RCG adsorption of phenol, and kinetic modeling showed pseudo-second-order model dynamics. From the physical-statistical modeling, it was found that the phenol adsorption for both cases is well represented by the monolayer model with two types of energy sites, irrespective of the origin of the clay. However, the characteristics of each material direct reflect the steric parameters, as indicated by the model. The RCA has a receptor site with low density but with a high number of molecules and another receptor site that presents a more equilibrate behavior, being able to increase the density with increasing system temperature. As for the RCG, it was found that both receptor sites had similar energy and tend to work together in the increment of the adsorption capacity. Potential thermodynamic functions indicate that the adsorption of phenol is spontaneous and endothermic for all the studied systems, with all the thermodynamic proprieties tending to increase in absolute values with increasing temperature. Furthermore, the thermodynamic analysis indicated that the RCG can further adsorb phenol since the equilibrium is not reached for any variable. Characterization of the solid after phenol adsorption confirmed the physical nature of the process and the type of layer (van der Waals) and the role played by hydrogen bonds in the case of RCG. The latter result suggests that the main interaction between the phenolics and the outer precursor is through electrostatic attraction.

**Supplementary Materials:** The following supporting information can be downloaded at: https://www.mdpi.com/article/10.3390/w15101881/s1.

**Author Contributions:** Y.D., D.S.P.F., J.G., T.L., R.O. and Y.B.: Conceptualization, Methodology, Survey, Original Draft Writing, Editing Review; Y.D., D.S.P.F., J.G., T.L., R.O., Y.B., H.M., H.O., A.S. and B.M.: Methodology, Survey, Original draft writing, editing; Y.D., D.S.P.F., Y.B., R.O. and T.L.: Conceptualization, Methodology, Resources, Writing, Editing, Supervision, Acquiring Funding. All authors have read and agreed to the published version of the manuscript.

**Funding:** This research received no external funding.

**Data Availability Statement:** Data sharing is not applicable to this article as no datasets were generated or analyzed during the current study.

**Acknowledgments:** The authors also thank everyone who helped prepare the manuscript.

**Conflicts of Interest:** We declare that we do not have any commercial or associative interest that represents a conflict of interest in connection with the work submitted.

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
