# Peer review of "Comparison of Phenol Adsorption Property and Mechanism onto Different Moroccan Clays"

_water, doi:10.3390/w15101881_

Round 1

Reviewer 1 Report

Reviewing Report

Titled: Title Unravel the mechanism of phenol elimination and com-2 pare the adsorption properties of phenol on two Moroccan clays from two different regions

Manuscript Number: water-2380691

The manuscript studied adsorption of phenol on two Moroccan  clays.  The authors used the physics-statistics modeling to describe the adsorption behavior. The topic is important and interesting. However, some unclear description needs to be clarified.  I suggested the manuscript should be revised before publication. The following comments could improve the manuscript.

Comments:

1.     The full name needs to be indicated, when abbreviation was used in the first time.

2.     The selected clays cannot generate high adsorptive amounts for phenol.  The authors need to describe the reason that they used the clays as adsorbents.  

3.     In Figure 5, the physical adsorption isotherms were defined as Type II, but not Type IV.

4.     For the TGA analysis, the authors need to explain the reasons of mass loss.

5.     Figure 7 is unclear. The authors need to rewrite it or replace it by other figure or table.

6.     Phenol might dissociate hydrogen ions when solution pH value is less than 5.5.  The authors need to discuss the potential effects on adsorption.

7.     The authors might increase description about phenol attached on the clays. The authors might accurately describe the adsorption behaviors.

The authors need to check the grammar.

Author Response

The manuscript studied adsorption of phenol on two Moroccan  clays.  The authors used the physics-statistics modeling to describe the adsorption behavior. The topic is important and interesting. However, some unclear description needs to be clarified.  I suggested the manuscript should be revised before publication. The following comments could improve the manuscript.

Comments:

  1. The full name needs to be indicated, when abbreviation was used in the first time.

Reply: Thank you very much for this remark see the corrected version

  1. The selected clays cannot generate high adsorptive amounts for phenol. The authors need to describe the reason that they used the clays as adsorbents. 

Reply: Thank you very much for this remark see the corrected version

  1. In Figure 5, the physical adsorption isotherms were defined as Type II, but not Type IV.

Reply: Thank you very much for this remark, from the analysis of the hystrsis loop and the comparison with the properties of the two types II and IV we conclude that type IV

Type II : solid completely covered by a monomolecular film (point B) then several layers of gaseous molecules (physisorption) Macroporous solids

Type IV : first follows a type II curve then plateau at high pressures, = filling of the pores then adsorption on the surface of the material. Mesoporous solids. desorption presents a hysteresis = condensation in the pores

  1. For the TGA analysis, the authors need to explain the reasons of mass loss.

Reply: Thank you very much for this remark see the corrected version

  1. Figure 7 is unclear. The authors need to rewrite it or replace it by other figure or table.

Reply: Thank you very much , the figure has been modified

  1. Phenol might dissociate hydrogen ions when solution pH value is less than 5.5. The authors need to discuss the potential effects on adsorption.

Reply: Thank you very much for this remark see the corrected version

  1. The authors might increase description about phenol attached on the clays. The authors might accurately describe the adsorption behaviors.

Reply: Thank you very much for this remark see the corrected version

Reviewer 2 Report

The article submitted for review in this version cannot be published. Apart from substantive issues (because the research presented is interesting and important), it needs to be organized so that a proper and objective assessment of the research carried out can be made. This version requires a major editorial revision. So I will list some of my observations:

1. Please remove the extra period from line 40

2. Please order the numbering of the chapters. This version of the article is unacceptable

3. Please complete the dots in lines 107 and 471, the entire article should be edited in terms of punctuation.

4. Please cite publications properly. And so, for example, in line 255, corrections must be made

5. There is no figure 2

6. I am asking for an editorial correction of the entire text. Section 3.8 needs correction

7. Please unify the titles

8. There is table 1 on page 16, previously there was table 1 on page 7. It needs to be sorted out. then there is table 5. I will repeat again that such a mess makes it difficult to properly assess the content of the research presented.

9. There are two chapters 10.4

10. After chapter 2.1 comes chapter 2.4

I am asking the authors to edit the text and send it for re-analysis.

Author Response

Report from Referee 2

The article submitted for review in this version cannot be published. Apart from substantive issues (because the research presented is interesting and important), it needs to be organized so that a proper and objective assessment of the research carried out can be made. This version requires a major editorial revision. So I will list some of my observations:

  1. Please remove the extra period from line 40

Reply: Thank you very much for this remark, the point has been deleted

  1. Please order the numbering of the chapters. This version of the article is unacceptable

Reply: Thank you very much for this remark see the corrected version

  1. Please complete the dots in lines 107 and 471, the entire article should be edited in terms of punctuation.

Reply: Thank you very much for this remark see the corrected version

  1. Please cite publications properly. And so, for example, in line 255, corrections must be made

Reply: Thank you , the form of the references corrected

  1. There is no figure 2

Reply: Thank you very much for this remark see the corrected version

  1. I am asking for an editorial correction of the entire text. Section 3.8 needs correction

Reply: Thank you very much for this remark see the corrected version

The cation exchange capacity was determined using the method of Delphine Aran [36]. Solids have a very high capacity compared to other types of clays [35].  The CEC of RCA is about 10.6 meq/100g, while that of RCG is 16.64 meq/100g. Compared with RCA, RCG has higher switching capacity. This difference is evidenced by the percentages of elements that make up the two materials.

  1. Please unify the titles

Reply: Thank you very much for this remark

  1. There is table 1 on page 16, previously there was table 1 on page 7. It needs to be sorted out. then there is table 5. I will repeat again that such a mess makes it difficult to properly assess the content of the research presented.

Reply: Thank you very much for this remark the numbering of the tables has been corrected

  1. There are two chapters 10.4

Reply: Thank you very much for this remark, the numbering of the titles has been checked

  1. After chapter 2.1 comes chapter 2.4

Reply: Thank you very much, the numbering of the titles has been checked

Round 2

Reviewer 2 Report

Dear Authors,

I would like to thank authors for careful and throughout reading of this manuscript and for the thoughtful correction, which help to improve the quality of this manuscript.

Dear Authors,

I would like to thank authors for careful and throughout reading of this manuscript and for the thoughtful correction, which help to improve the quality of this manuscript.

Author Response

Dear Editor,
First of all, thank you for giving us the opportunity to review our article titled
"Discovering the mechanism of phenol elimination and comparing the adsorption properties of phenol on two Moroccan clays from two different regions" (Manuscript number: No: water-2380691). The manuscript has been carefully revised following comments from the reviewers. We would like to thank the reviewers, as their comments have significantly improved the manuscript. All changes made in this new version are made by change tracking.
